# Biosensor Based Immunoassay: A New Approach for Serotyping of *Toxoplasma gondii*

**DOI:** 10.3390/nano11082065

**Published:** 2021-08-14

**Authors:** Susana Sousa, António Castro, José Manuel Correia da Costa, Eulália Pereira

**Affiliations:** 1LAQV, REQUIMTE, Department of Chemistry and Biochemistry, Faculty of Sciences, University of Porto, 4169 Porto, Portugal; 2Center for Parasite Biology and Immunology, National Institute of Health Dr. Ricardo Jorge, 4000 Porto, Portugal; antonio.castro@insa.min-saude.pt (A.C.); jose.costa@insa.min-saude.pt (J.M.C.d.C.); 3Center for the Study of Animal Science (CECA)/Institute for Agricultural and Agro-Alimentary Science and Technology (ICETA), University of Porto, 4051 Porto, Portugal

**Keywords:** biosensors, gold nanoparticles, *Toxoplasma gondii*, GRA6 antigen, polymorphic peptide

## Abstract

Toxoplasmosis is the most reported parasitic zoonosis in Europe, with implications in human health and in the veterinary field. There is an increasing need to develop serotyping of *Toxoplasma gondii* (*T. gondii*) in view of greater sensitivity and efficiency, through the definition of new targets and new methodologies. Nanotechnology is a promising approach, with impact in the development of point-of-care devices. The aim of this work was to develop a simple but highly efficient method for *Toxoplasma gondii* serotyping based on gold nanoparticles. A simple colorimetric method was developed using gold nanoparticles modified with the synthetic polymorphic peptide derived from GRA6 antigen specific for type II *T. gondii*. The method of preparation of the gold nanoprobes and the experimental conditions for the detection were found to be critical for a sensitive discrimination between positive and negative sera. The optimized method was used to detect antibodies anti-GRA6II both in mice and human serum samples. These results clearly demonstrate that a biosensor-based immunoassay using AuNPs conjugated with polymorphic synthetic peptides can be developed and used as a serotyping device

## 1. Introduction

Toxoplasmosis has the highest human incidence amongst the parasitic zoonosis [1]. The parasite, *Toxoplasma gondii* (*T. gondii*), is responsible for severe infectious and even death in newborns and in immunocompromised patients, and, eventually, immunocompetent patients [2,3]. These severe infections in immunocompetent patients are usually associated with virulent strains, generically classified as atypical. Several studies highlighted the association between strain genotype and pathogenesis [2,4]. In order to better understand the pathogenesis of toxoplasmosis, it is necessary to perform the genetic characterization (genotyping) of *Toxoplasma* associated with human and animal infections. Traditional approaches for genotyping are labor intensive and require the isolation of the infecting strain using a bioassay, or the isolation of parasite DNA. However, biological samples for bioassay are not always available. Moreover, the isolation of the parasite using bioassays may favor the isolation of one strain, while others may be missed in cases of mixed infections. To circumvent these problems, serotyping was proposed as an alternative method for the genetic characterization of *Toxoplasma* strains. Serotyping is a typing method based on the antibody recognition of strain-specific polymorphic peptides on serum samples [5,6,7,8] and does not require the isolation of the parasite. Furthermore, different antibodies produced against divergent strains will recognize strain-specific antigens, and thus facilitate the diagnosis of mixed infections. Serotyping was also proposed as an epidemiological tool for typing in areas where archetypal *T. gondii* infections prevail [9]. 

The development of rapid, cost-effective tests for the detection and identification of infectious pathogens has been widely pursued over the last decades. Rapid tests are usually seen as simple and inexpensive devices for qualitative or even quantitative biodetection that can be easily used not only by healthcare professionals, but also in the field. Nanotechnology holds much promise for the development of these rapid tests. In particular, gold nanoparticles (AuNPs) are particularly suitable for the development of rapid tests, not only due to their optical, magnetic and chemical properties [10] but also because of their biocompatibility, low cytotoxicity and immunogenicity, easy synthesis and functionalization [11]. Biosensors based in AuNPs have been developed for the sensitive detection of nucleic acids and proteins [12,13,14], including for the detection of antibodies for *T. gondii* [15]. They can become the next generation of diagnostic tools, as they show great sensitivity and specificity, replacing with advantage conventional molecular and serological methods [16]. AuNPs have been used for signal enhancement in order to improve the sensitivity and assay time of some classical methods, such as enzyme-linked immunosorbent assays (ELISA) [17]. In spite of the progress made thus far using nanotechnology to develop novel, simple and rapid diagnostic tests, only a few methods based on AuNPs have been reported for the detection of parasites [18,19,20,21,22,23,24,25]. To our knowledge, AuNPs have never been used in serotyping. The possibility of using AuNPs in a fast and easy assay to serotype *T. gondii* strains will provide the necessary data to better understand the link between strain genotype and pathogenesis [2,4] and to easily identify the virulent strains responsible for severe infections in immunocompetent patients.

In this work, we explored the optical properties of AuNPs to detect antibodies specific for the *T. gondii* GRA6 antigen, in order to demonstrate that an AuNPs-based biosensor can be an alternative to the classical serotyping methods such as ELISA. For that, a peptide that has polymorphisms specific for *T. gondii* genotype II was chosen. This synthetic peptide (here referred to as GRA6II), derived from the GRA6 protein C-terminal, is known to have good sensitivity and specificity on serotyping assays [7]. AuNPs were conjugated with peptide GRA6II to yield nanoprobes that specifically bind to anti-GRA6II antibodies. The binding of antibodies imparts different aggregation properties to the nanoprobes, enabling to easily discriminate between positive and negative samples (Figure 1), either by naked-eye detection or using UV/vis spectroscopy. The present work opens the possibility of developing rapid tests to be used in the serotyping of *Toxoplasma* infections. 

## 2. Materials and Methods

Chemicals and Reagents: Peptides GRA6II and CALNN were obtained from CASLO Laboratory (Lyngby, Denmark) and dissolved in deoxygenated water (Water for Molecular Biology, Sigma, St. Louis, MO, USA). Bovine Serum Albumin (BSA) protein standard and 11-mercaptoundecanoic acid (MUA) was purchased from Sigma Aldrich (St. Louis, MO, USA). All other chemicals and reagents were from Sigma Aldrich (St. Louis, MO, USA) and were of the highest purity available. Unless otherwise stated, all aqueous solutions were prepared with MilliQ water (18 MΩ cm). 

Gold Nanoparticle Synthesis: AuNPs with 28 nm and 42 nm were synthesized according to Bastús et al. [26]. AuNP concentration was determined using UV–Vis spectroscopy according to Haiss et al. [27]. Citrate 28-nanometer and 42-nanometer AuNPs were directly used for conjugation with GRA6II, or the citrate capping was replaced by MUA (on 42-nanometer AuNPs). For that, 10 mM MUA ethanolic solution was added to obtain a MUA:AuNPs ratio of 45.000. After overnight incubation, the AuNPs were washed by centrifugation at 3000× *g*, for 30 min, to remove free MUA in the solution. The pellet was resuspended in MilliQ water, and the solution was stored in the dark.

UV–Vis Spectrophotometry: UV–Vis spectrophotometry for characterization of AuNPs was performed in quartz cells with 1-centimeter path length (Hellma, Germany), using an GENESYS 10S UV–Vis spectrophotometer (Thermo Scientific) (Madison, WI, USA) UV–Vis spectrophotometry for aggregation ratio was performed using 96-well Nunc plates, using a Multiskan GO spectrophotometer (Thermo Scientific) (Madison, WI, USA). 

Dynamic Light Scattering and electrophoretic light scattering: DLS and ELS were performed in a Malvern Zetasizer Nano ZS (Malvern, Worcestershire, UK), at 25 °C. The following two types of bionanoconjugates at pH of 7.0, and 2 mM phosphate buffer were studied: 1) [AuNP–42 nm] = 0.25 nM and [GRA6II] of 0–700 µg/mL; and 2) [AuNP–28 nm] = 1 nM and [GRA6II] of 0–400 µg/mL. Each sample was measured at least 3 times; each measurement was the average of 100 sub-measurements. 

Preparation of GRA6II-BSA: Peptide GRA6II comprises the GRA6 C-terminal region between residues 220 and 230 (LHPGSVNEFDF). The peptide has only eleven amino acids and no cysteine residue is present. BSA was used as a carrier protein and coupled to GRA6II synthetic peptide. For that, 6 mg of GRA6II and 6 mg of BSA were dissolved in PBS, followed by the addition of 1 mL of glutaraldehyde 20 mM. After 1 h under stirring, the protein was left under dialysis at 4 °C in PBS for 48 h. The amount of GRA6II-BSA was then quantified using the Bradford assay at 595 nm, and stored in aliquots at −20 °C.

Serum samples: Serum samples from CD1 mice immunized with GRA6II-BSA were used as a positive control for the optimization of experimental conditions. Serum samples from CD1 mice negative for *T. gondii* were used as a negative control. ELISA was used to confirm that immunized mice were seropositive with antibodies anti-GRA6II. The maintenance and care of experimental animals complied with the Portuguese and European guidelines for the human use of laboratory animals. Mice were maintained in INSA animal facilities at the Centro de Saúde Pública Doutor Gonçalves Ferreira (CGF). The ethics committee for animal experimentation (ORBEA) approved all procedures according to License for Animal Experimentation approved on 28 June 2011. Mice were maintained in individual cages, and they received food and water ad libitum.

A pool of five GRA6II-positive serum samples from patients with Toxoplasma infection and a pool of five negative serum samples from patients without *Toxoplasma gondii* infection were used. These samples were collected in the scope of Toxoplasmosis screening program from Uruguay and retrospectively analyzed in a serotyping cross-sectional study [28]. The study was conducted on a group of anonymous human serum samples. 

AuNPs based immunoassay: Several experimental conditions were tested with mouse serum samples: sequential serum dilutions, [NaCl], [GRA6II-BSA] and pH. Briefly, citrate AuNPs were conjugated by overnight incubation at 4 °C, with GRA6II-BSA in phosphate buffer 2.0 mM at the following three pH values: pH 6.0, 7.0 and 8.0. The bionanoconjugates were washed twice with 2.0 mM phosphate buffer pH 7.0. The pellets were resuspended in 100 µl of serum sample dilutions ranging from 1:50 to 1:200 in 2.0 mM phosphate buffer pH 7.0 and left to incubate for 2 h. All the experiments were performed at room temperature (range 15–30 °C) and no significant changes were detected in assay results within this range of temperature. Finally, NaCl was added at 0.3 and 0.4 M and incubated overnight at room temperature. The following protocol established with mice immune sera was used, with minor modifications, in order to improve discrimination between positive and negative human samples: (i) AuNPs functionalization with MUA; (ii) a blocking step with 3% BSA in 2 mM phosphate buffer, pH 7.0; and (iii) [NaCl] = 0.5 M, in order to induce AuNPs aggregation. Each experimental condition was studied in triplicate.

Statistical analysis: Differences between arithmetic means were evaluated using Student’s *t*-test. Differences with a confidence interval of 95% or higher were considered statistically significant (*p* ≤ 0.05).

## 3. Results

### 3.1. Characterization of AuNPs

Spherical gold nanoparticles with an average size of 28 and 42 nm were chosen based on the different optical and aggregation properties. While 28-nanometer nanoparticles have a higher colloidal stability, and are easier to modify with biomolecules, 42-nanometer AuNPs have a more intense localized surface plasmon band, and thus may provide a better sensitivity to the assay. Both types of AuNPs were characterized using UV–Vis spectroscopy and DLS/ELS. The UV–Vis spectra show the localized surface plasmon resonance (LSPR) band with a maximum at 523 nm for 28-nanometer, and at 527 nm for 42-nanometer AuNPs (Appendix A). The hydrodynamic size obtained using DLS (Z-Average) was 28.0 ± 0.3 nm and 43.1 ± 0.5 nm, respectively. After the exchange of the capping agent used in the synthesis (citrate) for MUA, on 42-nanometer AuNPs, the hydrodynamic size slightly increased to 45.2 ± 0.9 nm. The zeta-potential value measured using ELS was −39.7 ± 1.4 mV, indicating a high colloidal stability.

### 3.2. Preparation and Characterization of AuNPs-GRA6II Probes

Three different capping agents for the AuNPs, namely 11-mercaptoundecanoic acid (MUA), thiolated polyethylenglycol (HS-(CH2)11-EG3-OCH2-COOH) and the pentapeptide CALNN, were assayed. For covalent conjugation, EDC/NHS was used [29]. In all these cases, either during the conjugation process, or for high concentrations of GRA6II, AuNPs became unstable and started to aggregate. For MUA-AuNPs, the variation of the zeta-potential with increasing concentrations of GRA6II was measured up to concentrations of 300 µg/mL, where extensive aggregation occurred (Appendix A). A small variation of the zeta-potential was observed, indicating that GRA6II does adsorb to AuNPs, forming stable conjugates up to a concentration of 200 µg/mL, where some aggregation can be already detected visually by the change in color of the AuNPs.

As an alternative approach, a conjugate of BSA with GRA6II (GRA6II-BSA) was incubated with citrate-AuNPs. The binding of GRA6II-BSA to AuNPs was studied using DLS and ELS, by adding increasing amounts of GRA6II-BSA (ranging from 10 to 400 µg/mL for 28-nanometer AuNPs and from 100 to 700 µg/mL for 42-nanometer AuNPs) to solutions of AuNPs and measuring the hydrodynamic diameter (Appendix A) [30] and zeta potential of the resulting conjugates (Appendix A). There is a significant increase in the hydrodynamic diameter and in the zeta potential with the concentration of GRA6II-BSA in a typical Langmuir adsorption curve (Appendix A). These results indicate that there is a strong binding of GRA6II-BSA to AuNPs, and that the resulting nanoprobes have a good colloidal stability. 

### 3.3. Colorimetric Immunoassay Using AuNPs-GRA6II-BSA

In order to optimize the experimental conditions to obtain a good discrimination between positive and negative samples, the following several factors were varied: (i) AuNPs’ size; (ii) the molar ratio between the antigen and the AuNPs; (iii) serum dilution; (iv) pH; and (v) the NaCl concentration. The best experimental conditions for discrimination were selected by comparison of the aggregation ratio measured as the absorbance ratio of positive and negative samples at two selected wavelengths. Selection of the two wavelengths was performed by subtracting the two experimental spectra and determining the maximum and minimum values. For both the 28-nanometer AuNPs and the 42-nanometer AuNPs, the selected wavelengths were 530 nm (non-aggregated) and 700 nm (aggregated). All the spectra were collected after an overnight incubation with NaCl.

### 3.4. 28-Nanometer AuNPs-GRA6II-BSA 

After the overnight incubation of AuNPs-GRA6II-BSA with positive mouse serum samples, a change of the AuNPs color from red to blue/purple was clearly detected. The UV–Vis spectra show a decrease in the plasmon resonance band at approximately 530 nm (non-aggregated) and the appearance of a new band at approximately 700 nm (aggregated), particularly when [NaCl] = 0.3 M was used (Appendix A). When comparing the aggregation ratio, the major difference between positive and negative serum aggregation ratio was with the dilution 1/200 when using [NaCl] = 0.3 M (Figure 2A). Those conditions were used for further studies. The following three GRA6II-BSA concentrations were used: 10, 40 and 100 µg/mL. The aggregation ratio for positive serum was higher when the bionanoconjugation was made with 100 µg/mL of GRA6II-BSA. The aggregation ratio difference between positive and negative sera was also higher for a protein concentration of 100 µg/mL (Figure 2B; see Appendix A). The pH at which AuNPs are conjugated with GRA6II-BSA is also an important factor. The following three different pH were used for the formation of bionanoconjugates: pH 6.0, pH 7.0 and pH 8.0. The bionanoconjugates formed under pH 7.0 showed a better difference between positive and negative aggregation ratio (Figure 2C; see Appendix A).

### 3.5. 42-Nanometer AuNPs-GRA6II-BSA 

Similar experimental conditions were tested with 42-nanometer AuNPs and GRA6II-BSA, except for the range of GRA6II-BSA concentration studied, which was increased by a factor corresponding to the increase in surface area of 42-nanometer AuNPs relative to 28-nanometer AuNPs, thus keeping a similar surface density of the GRA6II-BSA at both types of AuNPs. After an overnight incubation, there was a clear change of the AuNPs’ color between the positive and negative serum samples (see Appendix A). For these nanoparticles, the best discrimination between the positive and negative sera was observed with sera at 1/50 and [NaCl] = 0.3 M (Figure 3A). Regarding the optimal GRA6II-BSA concentration (100, 300 or 500 µg/mL), a higher aggregation ratio for positive serum was for 300 µg/mL. Furthermore, the difference between the positive and negative serum samples after the overnight incubation was also higher when the concentration of GRA6II-BSA was 300 µg/mL (Figure 3B). The AuNPs’ aggregation ratio was similar for the three pH conditions. The AuNPs-GRA6II-BSA conjugated at pH 7.0 had a higher aggregation ratio for both the positive and negative sample, when compared with pH 6.0 and pH 8.0. Furthermore, the major difference between the positive and negative serum was at pH 7.0 (Figure 3C). Similar experimental conditions were tested with AuNPs conjugated with BSA, in order to confirm the specific recognition of antibodies anti-GRA6II. No change of color was observed in the presence of both positive and negative serum samples with similar experimental conditions (Appendix A). 

### 3.6. 28-Nanometer AuNPs versus 42-Nanometer AuNPs

It is known that the size of spherical gold nanoparticles influences their ability to enhance the response of optical biosensors [31]. We compared the AuNPs’ aggregation ratio under the best experimental conditions for both sizes of AuNPs. Overall, the aggregation ratio of AuNPs-GRA6II-BSA in the presence of positive sera was higher for 28-nanometer AuNPs. However, the difference between the positive and negative sera was more intense with the larger AuNPs (Figure 4). 

### 3.7. Blocking of AuNPs-GRA6II-BSA Conjugates

In this study, the following blocking agents were tested: BSA, Tween 20 and ethanolamine. In the presence of Tween 20 and ethanolamine, both the positive and negative serum samples aggregated (Appendix A). These two agents clearly destabilize the bionanoconjugates. When BSA was used as the blocking agent, the difference between the aggregation ratio for the positive and negative samples was similar to the aggregation ratio in the absence of BSA (Figure 5). Based on this, we concluded that the blocking step with BSA is not necessary when GRA6II-BSA is used as an antigen, which indicates that the amount of GRA6II-BSA used in the bioconjugation process (300 µg/mL) is enough to create a complete monolayer and it is closed to the saturation point of the AuNPs. According to the zeta potential results, the saturation point for 42-nanometer AuNPs was obtained with a GRA6II-BSA concentration around 400 µg/mL (Appendix A). 

### 3.8. AuNPs-GRA6II-BSA Conjugates Stability

An important aspect on the development of an immunosensor is the stability of the bionanoconjugate. We analyzed the stability of the AuNPs-GRA6II-BSA conjugate during three weeks by measuring the surface charge using ELS. Surface charge is an important parameter to determine the stability and functionality of nanoconjugates. AuNPs-GRA6II-BSA zeta potential was measured at week one, two and three. A decrease was observed over time, indicating a loss of colloidal stability. The UV–Vis also shows a decrease in the concentration of AuNPs over time, possibly due to the sedimentation of larger aggregates (Appendix A). The AuNPs-GRA6II-BSA aggregation ratio during the three weeks increases from week one to week three for both the positive and negative samples (Figure 6). 

### 3.9. Optimized Colorimetric Immunoassay with Human Sera Pools

After optimization with mouse serum, a colorimetric immunoassay was tested with human sera pools from patients infected with genotype II *Toxoplasma* strains (positive sample) and without *Toxoplasma* infection (negative sample). 

The positive human sera pool was not able to induce a change of AuNPs-GRA6II-BSA color, in the same conditions used with mice immune serum. To overcome this limitation, peptide GRA6II was directly conjugated with AuNPs functionalized with MUA. We have chosen to assess these conjugates, since these are the easiest to prepare and the less expensive conjugates. The amount of peptide used was the maximum that allows the stability of the AuNPs. Amounts superior to 200 µg/mL induce the aggregation of the AuNPs. With this amount, the colloidal stability of the AuNPs is preserved. A blocking step with BSA was introduced to avoid unspecific antibody binding to the AuNPs-GRA6II, increasing the specificity of the interaction between the conjugates and the antibodies. 

An appropriate salt concentration is essential for the immunoassay, since less salt will affect the sensitivity and more salt will greatly reduce its capability against interference. Thus, a salt concentration of 0.5 M was chosen. Lower salt concentrations did not induce aggregation of positive samples, while with higher concentrations, conjugates became unstable with time (Figure 7). 

Under the described experimental conditions, and after overnight incubation, there was a clear difference of the AuNPs’ UV–Vis spectrum (Figure 7A) and color between the positive and negative serum samples. MUA-AuNPs GRA6II turned blue in the presence of positive human sera pool. On the contrary, MUA-AuNPs in the presence of negative human sera pool remained red (Figure 1B). 

## 4. Discussion

The aim of this study was to develop an immunoassay for the detection of antibodies anti-*Toxoplasma gondii* specific for GRA6II in serum samples. The immunoassay was based on the change of AuNPs’ color upon aggregation. AuNPs were conjugated with a synthetic peptide, derived from *Toxoplasma* GRA6 antigen. This peptide has polymorphisms specific for genotype II and showed a good sensitivity and specificity on serotyping assays [5,7]. It is, thus, expected that antibodies anti-GRA6 antigen will specifically recognize the GRA6II peptide conjugated to AuNPs. 

One of the major factors that can affect the stability of the nanoprobes, and thus the efficiency of nanoprobe-based bioassays, is the type of conjugation used to bind the recognition molecule, in our case GRA6II peptides, to AuNPs. The following two major approaches are usually performed: non-covalent and covalent conjugation. Non-covalent conjugation is easier to perform and less expensive than covalent conjugation, thus having the potential to decrease the final cost of the test. In addition, non-covalent conjugation is known to induce less conformational changes in the bound biomolecule, thus preserving its binding properties to the biological partner. However, covalent conjugation usually provides nanoprobes that are more resistant to degradation in biological media. As an alternative approach to improve the binding of GRA6II to AuNPs, GRA6II-BSA was incubated with citrate-AuNPs. This approach takes advantage of the strong binding of BSA to citrate AuNPs by electrostatic interactions [32] and possibly also by chemisorption through many cysteine residues in the protein. BSA conjugates with AuNPs show high colloidal stability against salt, acid and base addition [33]. Moreover, GRA6II-BSA was used as an antigen to produce mouse immunosera; therefore, it is expected to have a stronger affinity of these probes to the mouse antibodies in immunosera used in the study. This strategy was found to greatly improve not only binding to AuNPs, but also the colloidal stability of the resulting nanoprobes.

AuNPs functionalized with MUA and conjugated with GRA6II proved to be more efficient when human serum samples were used. This may be explained by the fact that when BSA was used as a carrier protein for the conjugation of GRA6II to AuNPs, the concentration of protein added to the AuNPs did not reflect the real concentration of GRA6II. In fact, the real concentration of GRA6II is lower than 300 µg/mL. The mice immune sera used for the optimization of experimental conditions has specific anti-GRA6II-BSA antibodies. On the contrary, the pool of positive human sera has anti-*Toxoplasma gondii* antibodies, meaning that it has antibodies against a wide range of *Toxoplasma* antigens, including the GRA6II antigen [28].

A fundamental step in every immunodiagnosis assay is blocking. The objective of this step is to eliminate the unspecific binding of serum antibodies to the uncoated spots of AuNPs that could lead to false positive reactions. The optimal blocking agent for any particular assay must be determined by empirical testing [34]. A blocking step was not considered important when conjugate AuNPs-GRA6II-BSA was used. However, with AuNPs-MUA, a blocking step with BSA was introduced to minimize the non-specific interactions between the AuNPs and the serum proteins, increasing the specificity of the interaction between the conjugates and the antibodies. BSA is a protein routinely used for this purpose in immunoassays [35]. The blocking step in immunoassays based on conjugates AuNPs-antigens is not considered crucial for all AuNPs-antigens conjugates [35]. However, it was maintained in order to avoid non-specific interactions with human sera components.

The present assay discriminates between GRA6II-AuNPs with and without bound antibodies due to their different colloidal stability of the GRA6II-AuNPs conjugates. The antibody–antigen interaction changes the surface properties of GRA6II-AuNPs, in particular their surface charge and colloidal stability. The surface charge of AuNPs bioconjugates reflects the conformation of biomolecules on the nanoparticles, affects the interactions between particles and biological molecules and tunes the stability of nanobioconjugates [30]. 

Immunosensors with colorimetric properties are considered the future of diagnosis immuno devices [16]. This work optimizes the experimental conditions for an immunosensor with the potential to be used on *Toxoplasma gondii* serotyping assays. Previous serotyping studies are based on the use of polymorphic synthetic peptides [5,7] or recombinant peptides [6] as the target antigen in ELISA and peptide microarray assays [36]. The innovative character of our study stems from the use of a synthetic polymorphic peptide in association with gold nanoparticles to design an alternative method for the serotyping of *Toxoplasma gondii*. The AuNPs-based method shows a Limit of Detection (LOD) [37] of 0.481, while for ELISA it is 0.979. In spite of the limited number of human serum samples, these values suggest the high sensitivity of this method when compared with the classical ELISA approach. The strain type has been suggested to play a role in determining the outcome of *Toxoplasma gondii* infection [9]. The development of an easy colorimetric test for serotyping *Toxoplasma gondii* should make it possible to substantially expand the number of patients and the range of disease in which the strain type can be determined and should, eventually, help to understand how strain type influences disease outcome, and for a more accurate treatment of infections. A rapid and easy colorimetric test will also be of major importance in the veterinary field. It is known that *Toxoplasma* infection in sheep may result in foetus loss with the consequent negative economic impact in animal production. Previous serotyping studies have shown that peptide GRA6II is able to interact specifically with anti-*Toxoplasma gondii* IgG antibodies in serum samples from patients with natural infections [5,7]. This peptide, however, demonstrates some limitations for typing strains of animal origin [38]. One of the challenges of immunoassays, such as ELISA and animal serum samples from natural infections, is the use of an appropriate secondary antibody. The serotyping strategy proposed here overcomes this limitation, since no secondary antibody is necessary. Finally, it has the potential to become a routine analytical method for the typing of lineages for epidemiological studies to screen the lineages present in a given population.

This study provides a proof-of-concept that a biosensor-based immunoassay using AuNPs conjugated with polymorphic synthetic peptides can be developed and used as a serotyping device. A study to validate this approach in order to assess the sensitivity, specificity and accuracy of this and other nanoconjugates is ongoing. 

## Figures and Tables

**Figure 1 nanomaterials-11-02065-f001:**
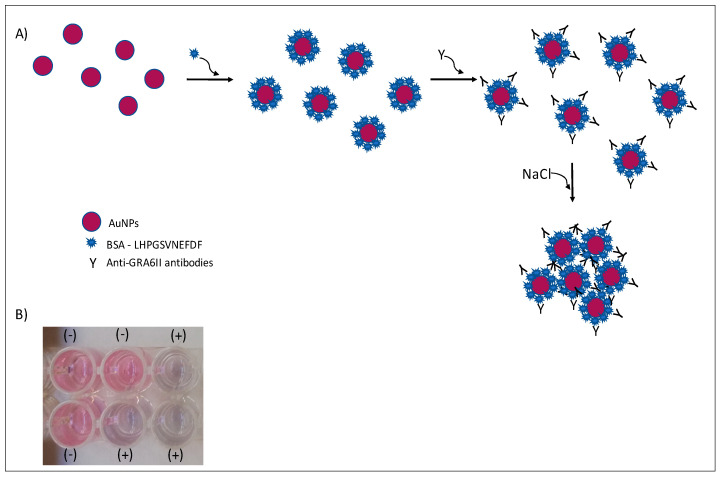
Colorimetric immunosensor by aggregation of AuNPs. In the presence of specific antibodies and salt, AuNPs will aggregate (**A**). A change in color is observed in positive (+) serum samples (**B**).

**Figure 2 nanomaterials-11-02065-f002:**
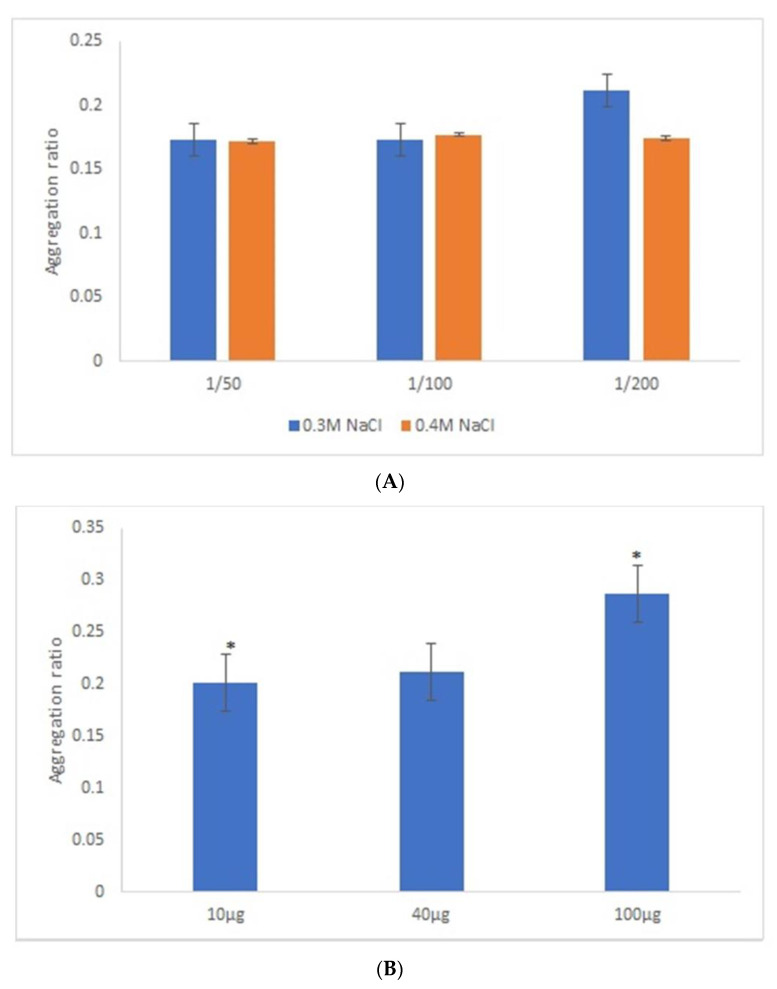
Aggregation ratio variation between positive and negative serum samples for 28-nanometer AuNPs bioconjugated with GRA6II-BSA. (**A**) Three sera dilutions were tested (1/50, 1/100 and 1/200) and two NaCl concentrations (0.3 and 0.4 M). (**B**) The following three GRA6II-BSA concentrations were compared: 10, 40 and 100 µg/mL. (**C**) The following three pH values were compared: pH 6.0, pH 7.0 and pH 8.0. * represents statistically significant differences between aggregation ratio of positive and negative serum samples, for each experimental condition.

**Figure 3 nanomaterials-11-02065-f003:**
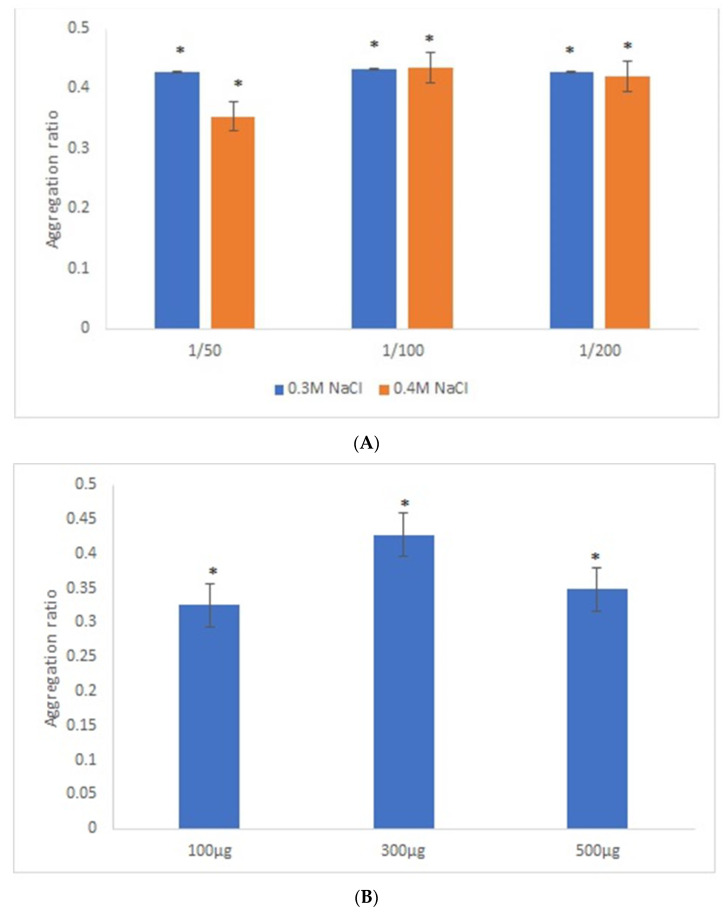
Aggregation ratio variation between positive and negative serum samples for 42-nanometer AuNPs bioconjugated with GRA6II-BSA. (**A**) Three sera dilutions were tested (1/50, 1/100 and 1/200) and two NaCl concentrations (0.3 and 0.4 M). (**B**) The following three GRA6II-BSA concentrations were compared: 100, 300 and 500 µg/mL. (**C**) The following three pH values were compared: pH 6.0, pH 7.0 and pH 8.0. * represents statistically significant differences between aggregation ratio of positive and negative serum samples, for each experimental condition.

**Figure 4 nanomaterials-11-02065-f004:**
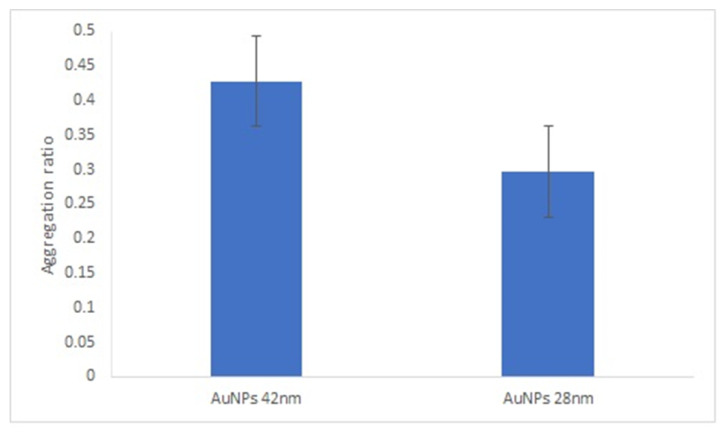
Effect of AuNPs size on AuNPs bioconjugated with GRA6II-BSA. Aggregation ratio variation between positive and negative serum sample.

**Figure 5 nanomaterials-11-02065-f005:**
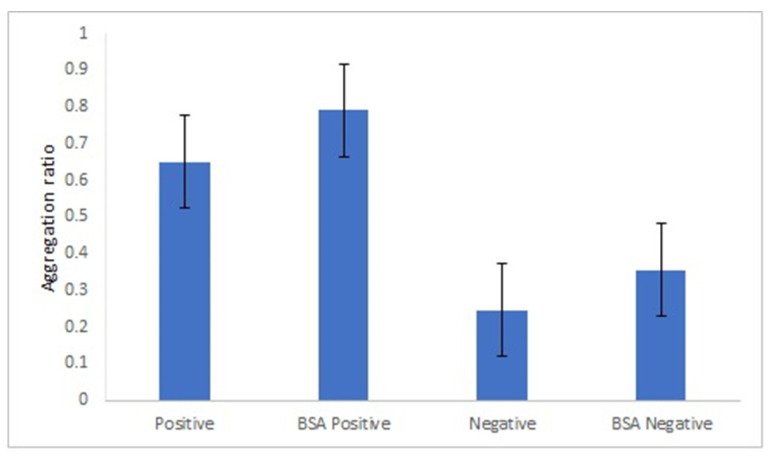
Effect of BSA blocking step in the aggregation ratio of 42-nanometer AuNPs GRA6II-BSA.

**Figure 6 nanomaterials-11-02065-f006:**
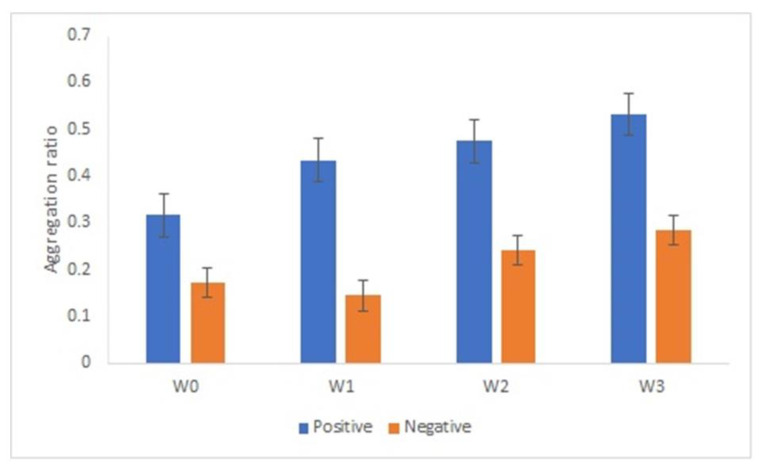
AuNPs’ aggregation ratio in the presence of positive and negative serum samples after 1 week (W1), 2 weeks (W2) and 3 weeks (W3) of storage at 4 °C.

**Figure 7 nanomaterials-11-02065-f007:**
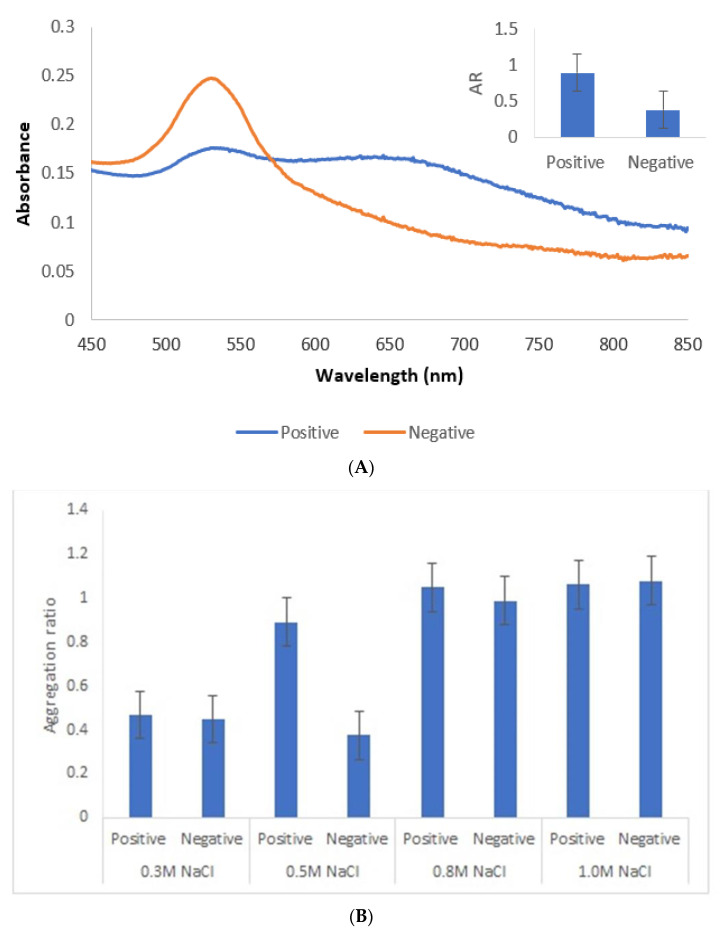
Immunoassay with human sera pool from patients infected with genotype II strains (positive) and without *Toxoplasma* infection (negative). (**A**) UV–Vis spectrum and AuNPs aggregation ratio (AR) for positive and negative serum samples. (**B**) AuNPs Aggregation ratio for positive and negative sera pool under different concentrations of NaCl.

## Data Availability

The data presented in this study are available in Appendix A.

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
