# Peer review of "Biosensor Based Immunoassay: A New Approach for Serotyping of Toxoplasma gondii"

_nanomaterials, 2021, doi:10.3390/nano11082065_

Round 1

Reviewer 1 Report

The Ms. report a first proof of principle work aimed at the development of a nanostructured sensor for Toxoplasma gondii based on a colorimetric assay. The topic is relevant for the research community. The method is not new but there is the effort of the Authors to carefully determine the physical and chemical conditions/parameters  that maximize the sensitivity. 

The Ms. is well written and the results as they are presented are clear. However, I have some major concerns that in my opinion should be addressed before publications. 

  1. The Authors explore covalent and non-covalent binding strategies. They report extensively the aggregation conditions for non-covalent binding based on the electrostatically driven BSA-GRA6 antigen / citrate interaction. However, these types of constructs do not work for human sera and they resolve to use the MUA-antigen binding strategy. It seems to me that more extensive characterization should be given to the MUA-GRA6 construct with respect to the others that, though working with mice sera, do not give enough sensitivity for human sera. In particular, it is recommended that the Authors smooth the statement "Only a small variation was detected indicating poor binding of the peptide to the AuNPs. " referring to MUA based constructs, and better substantiate their choice of this construct in the second part of the work.
  2. Correlated to this, I do not see an attempt to quantify the sensitivity in terms of the Limit Of Detection. LOD can be given in-vitro only with purified proteins. The Authors correctly refer to the sera for testing. In this case one should analyze the sera (positive and negative ones) with some alternative quantitative testing (if it exists) to provide a comparison. 
  3. The colorimetric assay is built on the ratio or difference of the absorbance at two wavelengths. In the methods the Authors write that these have been chosen so to maximize the signal but do not provide the values. The doubt is that that chose these two values differently for different experimental conditions, a fact that would increase the variability of the results.  It is important that the Authors clarify this point.
  4. In Figs.2 and 3, at least, a T-test should be provided to understand to what extent the variation in the response between different conditions is relevant.

Minor points.

  1. The data are mainly provided as "aggregation rate". I miss the point of the Ms. where the units for the aggregation rate are given. 
  2. In Fig.2 and Fig.3 there is a difference in the range of GRA6 peptide used to coat the NPs. Is this due only to the increased surface of the NPs? In Fig4 the comparison is made at the same concentration of peptide? Which one?
  3. A minor typo is "essay" instead of "assay". 

Reviewer 2 Report

This manuscript provides a colorimetric method using gold nanoparticles modified with the synthetic polymorphic peptide derived from GRA6 antigen specific for type II Toxoplasma gondii. Different experimental conditions for the detection were found to have different influence for discrimination between positive and negative sera. Current study provides a biosensor-based immunoassay using AuNPs conjugated with polymorphic synthetic peptides can be developed and used as a serotyping device. In general, the data provided in this study can support its conclusion. But I still have a few major concerns:

  1. The application of gold nanoparticle for Toxoplasma gondii serotyping using immunoassay has been long studied and reported in several papers such as 10.1016/j.sbsr.2018.05.002, 10.1016/S0956-5663(03)00265-3. What is the special novelty and superiority of current study?
  2. Characterization of gold nanoparticle is not sufficient. For instance, high quality SEM should be provided.
  3. The expression of GRA6II should be measured.
  4. Salt concentration was measured only using 0.3 and 0.4M in figure 2 and 3, but different concentration ladder was used for figure 7. Why is that?
  5. Has the author considered the influence of temperature.
  6. Why size of 42 and 28nm were chosen?
  7. No statistical analysis is provided.
  8. In figure 7b, salt concentration was indicated using comma instead of dot (0,3 instead of 0.3), please revise.

Round 2

Reviewer 1 Report

The Authors have answered to all my concerns in a satisfactory way but for two specific points for which I suggest to take a closer look.
1. in the point 3 of the answers they cite their sentence " Selection of the two wavelengths was performed by subtracting the two experimental spectra and determining the maximum and minimum values. For 28 nm AuNPs the selected wavelengths were 530 nm (non-aggregated) and 700 nm (aggregated) and for 42 nm AuNPs were 530 nm (non-aggregated) and 700 nm (aggregated). All spectra were collected after overnight incubation with NaCl." It seems weird that, basing on what they correctly write in the answer, the very same lambdas are used for the 2 types of NPs. I would expect a slightly different change in the shorter lambda. However, if it so, maybe the sentence could be changed to "Selection of the two wavelengths was performed by subtracting the two experimental spectra and determining the maximum and minimum values. For both the 28 nm AuNPs and the 42 nm AUNPs the selected wavelengths were 530 nm (non-aggregated) and 700 nm (aggregated) . All spectra were collected after overnight incubation with NaCl."

2. The T test: as far as I now it should be done between two distributions. So, in Fig.2A can be given for each of the couples of bars blue and orange and in Fig.2b it si not clear between which distributions the T test is done (probably the left most and the right most bars?). Similar considerations can be given for the other figures. I ask the Authors to check for this.

Reviewer 2 Report

Temperature can vary much depending on season, air-con and so on. It is recommended to study the temperature influence in this work, which should be as important as pH and many other influencing factors. 
